# Computation Offloading in UAV-Enabled Edge Computing: A Stackelberg Game Approach

**DOI:** 10.3390/s22103854

**Published:** 2022-05-19

**Authors:** Xinwang Yuan, Zhidong Xie, Xin Tan

**Affiliations:** National Innovation Institute of Defense Technology, Academy of Military Science of the People’s Liberation Army, Beijing 100071, China; alter_yxw@163.com (X.Y.); tanxin2017@163.com (X.T.)

**Keywords:** mobile edge computing, unmanned aerial vehicles, computation offloading, Stackelberg game

## Abstract

This paper studies an efficient computing resource offloading mechanism for UAV-enabled edge computing. According to the interests of three different roles: base station, UAV, and user, we comprehensively consider the factors such as time delay, operation, and transmission energy consumption in a multi-layer game to improve the overall system performance. Firstly, we construct a Stackelberg multi-layer game model to get the appropriate resource pricing and computing offload allocation strategies through iterations. Base stations and UAVs are the leaders, and users are the followers. Then, we analyze the equilibrium states of the Stackelberg game and prove that the equilibrium state of the game exists and is unique. Finally, the algorithm’s feasibility is verified by simulation, and compared with the benchmark strategy, the Stackelberg game algorithm (SGA) has certain superiority and robustness.

## 1. Introduction

With the rapid development of network communication technology, the data interaction efficiency of the mobile Internet is constantly improving. Meanwhile, the transmission bandwidth and data scale have become increasingly large. 5G communication and cloud computing have spawned new applications such as driverless, automatic navigation, face recognition, and augmented reality [1]. Meanwhile, these applications are computationally intensive and time-sensitive. However, mobile devices at the terminal cannot provide sufficiently high-performance computing services, and the battery capacity is limited, so it is inefficient in handling these tasks and may not meet the quality of services requirements [2]. In the network architecture of cloud computing, computing resources concentrate in the cloud, and there is a certain distance between the computing resources and terminal devices. Therefore, the service response has an inevitable delay [3], and when dealing with computationally intensive tasks; it is prone to access congestion.

Mobile edge computing (MEC) is a new computing architecture for providing computing services [4,5] that can push the service resources of cloud computing to the edge to meet the requirements of intensive computing and low latency. Compared with cloud computing, edge computing is more in line with the concept of a smart city, which is proposed to realize green and sustainable development [6]. Traditional terrestrial networks face challenges in scenarios such as complex terrain and equipment failure. Unmanned aerial vehicles (UAV) can help to enhance the flexibility and robustness of mobile edge computing network deployment [7], and reduce the complexity and cost of resource management. For example, Verizon and AWS work together to combine UAVs with mobile edge computing to reduce connection latency and reduce UAV costs by about 10% [8]. However, with the increasing number of UAVs in use, resource management of networks faces challenges such as power control, spectrum allocation, interference management, and task allocation [9].

In terms of task allocation, different devices in the MEC network differ in various aspects, such as onboard battery capacity and computational performance. Tasks that require high performance can be offloaded to devices with large battery capacity and high computing performance. In MEC networks, the offloading forms of computation include partial offloading (complex computation tasks can be divided into several sub-computation tasks.) and full offloading (tasks are inseparable) [10]. The computation task offload strategy should consider determining the proportion of the offloading task and choosing the offloading computation content [11]. Moreover, the balance between throughput and fairness also needs to be considered [12]. Artificial intelligence (AI) could also help to allocate resources dynamically. Combined with AI, the learning capacity of edge devices could be enhanced [13]. As shown in Figure 1, the UAV in the MEC network can act as a user at the terminal to access the MEC service or a server to receive the tasks from users; it can also act as a relay node to forward tasks offloaded from nearby users to the base station servers. Roles are changed according to the scenarios.

### 1.1. Related Works

To improve the performance of the computational tasks’ allocation strategy, it is necessary to set the optimization goal first. On the one hand, some studies considered a single performance metric. Long et al. [14] aimed to minimize the computation latency and obtain the optimal offloading computation task strategy. However, it is necessary to consider introducing a UAV-enabled MEC network to improve the performance of computing offload further. Luan et al. [15] aimed to minimize the energy consumption and solve the problem of distributed task allocation of UAVs based on a coalitional game. On the other hand, some studies have integrated multiple performance metrics, which could improve the system computing performance more comprehensively. Chen et al. [16] integrated computing latency and energy consumption, and then introduced the pricing of computing resources to differentiate the servers on base stations and UAVs, but it is also necessary to analyze the allocation scheme of the optimal offloading proportion. Ren et al. [17] considered to minimize the global computation latency and energy consumption and divided the computation task into several subtasks, then worked out the optimal offloading proportion by KKT-condition. In addition, in that study, the optimal offloading strategy and minimum computing latency were obtained by a non-cooperative game. However, due to the limited endurance of UAVs, a single UAV cannot provide services for ground users efficiently, so the performance of multiple UAVs should also be considered.

In addition, some studies have considered the resource allocation in the more complex network environment. Alioua. et al. [18] comprehensively considered latency, energy consumption, link quality, and other factors in a multi-UAV-enabled road traffic monitoring scenario, and obtained a computation resource allocation strategy with better overall performance based on the sequential game. EI N.N et al. [19] considered device association, task assignment, and computational resource allocation problems comprehensively, and proposed an iterative algorithm based on block coordinate descent to minimize the energy consumption of mobile devices and UAVs. Yan et al. [20] considered the MEC network of a multi-UAV and multi-ground user scenario and constructed a Stackelberg game model: The UAV, as the vice-leader, maximizes the income by optimizing the location and pricing of computing resources. In addition, the user, as the follower, reduces the overhead by optimizing the offload allocation strategy. Dong et al. [21] points out that most studies focus on simple application scenarios, and the research on multi-UAV cooperative resource management in complex environments needs to be further deepened and improved.

### 1.2. Contributions

According to related works above and [22,23,24,25,26], game theory has a good application prospect in solving resource management of the MEC network recently. In addition, we found that multiple UAV applications should be considered in MEC resource management, and energy consumption, delay, and other aspects should be considered comprehensively in performance indexes to improve system performance comprehensively. In Stackelberg game, the player can decide the strategy at different layers. The followers respond to the leader’s strategic actions and choose the best strategy. Therefore, players at all levels can consider their own interests and set different utility functions. Previous studies [27,28] have used the Stackelberg game to solve the problem of computing unloading in MEC, we extend it to the UAV-enabled application scenario, and it is different that we divide players into three layers based on the supply of computation. The contributions of this study are as follows:In the multi-UAV cooperative MEC network, we construct a multi-layer Stackelberg game model according to the different characteristics of base stations, UAVs, and users. And the equilibrium state is reached through several iterations to meet the maximum interests of players in each layer.Players at each layer of the Stackelberg game set the utility function according to their own interests and demands, and select the optimal computational offloading strategy to maximize the total utility of the system.Performance metrics such as computation latency and energy consumption are weighted to improve the performance of the system more comprehensively.The proposed SGA has a certain superiority and robustness. Compared with the benchmark scheme of random pricing strategy, it can achieve higher total system benefits in multiple different scenarios.

## 2. Model of Stackelberg Game

The main symbols in the model are given in Nomenclature.

### 2.1. Leader Sub-Game

In the MEC network, the ground base station has strong computing ability and continuous power supply, so the base stations act as the leader and the UAV as the vice-leader in the Stackelberg game. For each leader station *g*, the onboard edge server can receive computing tasks directly from the user or employ the UAVs as relay nodes to forward computing tasks from the users. The leader needs to set a different price $M_{i}^{g}$ for each computing resource requested by user *i*, a corresponding price $M_{j}^{g}$ for the employed relay UAV *j*. The computing resource $F_{g}$ of the leader is limited, and the computing resource allocated to the user *i* is $F_{i}^{g}$. Then we can obtain the profit $P_{g}$ that the base station by providing computing services to the user and the cost $C_{g}$ of employing the UAV as follows:(1)$$P_{g}=\sum_{i\in I}M_{i}^{g}F_{i}^{g}Need_{i}^{g}$$
(2)$$C_{g}=\sum_{i\in I}\sum_{j\in J}M_{j}^{g}D_{i}Need_{i}^{j}\gamma_{i}^{r}$$
where $D_{i}$ represents the computation task quantity of user *i*, $Need_{i}^{g}=1$ indicates that user *i* needs to offload the computing task to base station g, $Need_{i}^{j}=0$ indicates that the computing task of user *i* needs the assistance of UAV *j*, and $\gamma_{i}^{r}$ indicates the proportion that user *i* chooses to forward the computing task to the base station through relay.

Then, the net income of the leader base station *g* can be expressed as $U_{g}=P_{g}-C_{g}$, and the game of the leader layer can be formulated as:(3)$$\max U_{g}\left(M_{i}^{g},M_{j}^{g},F_{i}^{g}\right),\forall i\in I,j\in J$$
(4)$$s.t.\left\{\begin{array}{cc} \sum_{i\in I}F_{i}^{g}\leq F_{g} & F_{i}^{g}\leq D_{i}\\ M_{i}^{g}\in \left[\min M_{i}^{g},\max M_{i}^{g}\right]\\ M_{j}^{g}\in \left[\min M_{j}^{g},\max M_{j}^{g}\right]\\ j_{g}⋂j_{g^{\prime }}=ø & \forall g,g^{\prime }\in G \end{array}\right.$$
where $M_{i}^{g}$ is the pricing set of all base stations for user *i*, $M_{j}^{g}$ is the base station’s pricing set for hiring UAVs, $F_{i}^{g}$ is the resource set allocated to the user by the base station, I,J,G are respectively the set of users, UAVs, and base stations. The first constraint indicates that the computing resources of the base station are limited, the second and third constraints indicate that the pricing for user services and hiring UAVs should fall within a certain range, and the fourth constraint indicates that each UAV can only be employed by one base station.

### 2.2. Vice-Leader Sub-Game

In the MEC network, the UAV can act as a server to receive computing service requests from users or as a relay node to forward service requests from users to the base station. $Role_{i}^{j}=1$ represents that the UAV *j* acts as a server, and $Role_{i}^{j}=0$ represents a relay node when processing the computing tasks of user *i*. The benefits of these two roles are denoted as $P_{j}^{compute}$ and $P_{j}^{relay}$, respectively. When acting as a server, the UAV sets a price $m_{i}^{j}$ for the computing resources accessed by different users. As with the BS in the leader layer, the service resource $F_{j}$ of the UAV is also limited, and the resource allocated to user *i* is denoted as $f_{i}^{j}$. Then the profit $P_{j}^{compute}$ and $P_{j}^{relay}$ of the vice-leader can be expressed as:(5)$$P_{j}^{compute}=\sum_{i\in I}m_{i}^{j}f_{i}^{j}Role_{i}^{j}$$
(6)$$P_{j}^{relay}=\sum_{i\in I}\sum_{j\in J}M_{j}^{g}D_{i}Need_{i}^{j}\left(1-Role_{i}^{j}\right)$$

When UAV *j* acts as an edge server, there is overhead $C_{j}^{compute}$ in processing computation tasks, and when acting as a relay, there is the communication cost $C_{j}^{trans}$ in transmitting the tasks:(7)$$C_{j}^{compute}=\sum_{i\in I}\frac{\varphi_{i}}{f_{i}^{j}}\eta_{i}Need_{i}^{j}Role_{i}^{j}$$
(8)$$C_{j}^{trans}=\sum_{i\in I}\sum_{j\in I}p_{j}\frac{D_{i}}{rate_{i}^{j,g}}Need_{i}^{j}\left(1-Role_{i}^{j}\right)$$

Therefore, the profit of the vice-leader UAV *j* can be expressed as $U_{j}=P_{j}^{compute}+P_{j}^{relay}-C_{j}^{compote}-C_{j}^{trans}$, and the game of the vice-leader can be formulated as:(9)$$\max U_{j}\left(m_{i}^{j},f_{i}^{j}\right),\forall i\in I$$
(10)$$s.t.\left\{\begin{array}{c} Role_{i}^{j}\in \left\{0;1\right\}\\ m_{i}^{j}\in \left[\min m_{i}^{j},\max m_{i}^{j}\right]\\ \sum_{i\in I}f_{i}^{j}\leq F_{j},f_{i}^{j}\leq D_{i} \end{array}\right.$$

### 2.3. Follower Sub-Game

As a follower, user *i* taking the fees $C_{i}^{pay}$ paid for computation services and the processing cost $C_{i}^{compute}$ (including the computation latency and communication cost in transmission) into account and decides the offloading object and proportion based on the pricing strategy given in the leader layer and the allocated computing resources.
(11)$$C_{i}^{pay}=\left\{\begin{array}{cc} 0, & \gamma_{i}^{l}\geq 0\\ \alpha_{g,i}^{pay}M_{i}^{g}F_{i}^{g}, & \gamma_{i}^{g}\geq 0\\ \alpha_{j,i}^{pay}m_{i}^{j}f_{i}^{j}, & \gamma_{i}^{j}\geq 0\\ \alpha_{relay,i}^{pay}M_{i}^{g}F_{i}^{g}, & \gamma_{i}^{r}\geq 0 \end{array}\right.$$
(12)$$C_{i}^{compute}=\left\{\begin{array}{cc} \alpha_{local,i}^{trans}\frac{\varphi_{i}}{f_{i}}+\alpha_{local,i}^{exe}\frac{\varphi_{i}}{f_{i}}\eta_{i}, & \gamma_{i}^{l}\geq 0\\ \alpha_{g,i}^{time}\left(\frac{D_{i}}{Rate_{i}^{g}}+\frac{\varphi_{i}}{F_{i}^{g}}\right)+\alpha_{g,i}^{energy}p_{i}\frac{D_{i}}{F_{i}^{g}}, & \gamma_{i}^{g}\geq 0\\ \alpha_{j,i}^{time}\left(\frac{D_{i}}{Rate_{i}^{j}}+\frac{\varphi_{i}}{f_{i}^{j}}\right)+\alpha_{j,i}^{energy}p_{i}\frac{D_{i}}{f_{i}^{j}}, & \gamma_{i}^{j}\geq 0\\ \alpha_{relay,i}^{time}\left(\frac{D_{i}}{Rate_{i}^{j}}+\frac{\varphi_{i}}{F_{i}^{j}}+\frac{D_{i}}{Rate_{i}^{j,g}}\right)+\alpha_{relay,i}^{energy}p_{i}\frac{D_{i}}{Rate_{i}^{j}}, & \gamma_{i}^{r}\geq 0 \end{array}\right.$$
where $p_{i}$ denotes the user’s transmission power, and $\gamma_{i}^{l}$, $\gamma_{i}^{g}$, $\gamma_{i}^{j}$ and $\gamma_{i}^{r}$ denote the allocation ratios for local computing, offloading to the base station, offloading to the UAV and forwarding to the base station via relay, respectively. Then, the utility function of user *i* can be expressed as: $U_{i}=-C_{i}^{compute}-C_{i}^{pay}$, and the game of the follower can be defined as:(13)$$\max U_{i}\left(\gamma_{i}^{l},\gamma_{i}^{g},\gamma_{i}^{j},\gamma_{i}^{r}\right)$$
(14)$$s.t.\left\{\begin{array}{c} \gamma_{i}^{l},\gamma_{i}^{g},\gamma_{i}^{j},\gamma_{i}^{r}\in \left[0,1\right]\\ \gamma_{i}^{l}+\gamma_{i}^{g}+\gamma_{i}^{j}+\gamma_{i}^{r}=1 \end{array}\right.$$

### 2.4. Computation Offload Model Based on Stackelberg Game

The game of base station, UAV, and user from the first three sections can form a three-layer heterogeneous Stackelberg game model, which can be expressed as:(15)$$G=\left\{\left(G,J,I\right),\left(G,J,I\right),\left(U_{g},U_{j},U_{i}\right)\right\}$$
where G,J,I are the sets of strategy space, respectively. For this Stackelberg game model, to get the state of Stackelberg Equilibrium(SE), the set $\left(M_{i}^{g*},M_{j}^{g*},m_{i}^{j*},F_{i}^{g*},f_{i}^{j*},\gamma_{i}^{l*},\gamma_{i}^{g*},\gamma_{i}^{j*},\gamma_{i}^{r*}\right)$ should satisfied the conditions as follow:(16)$$U_{g}\left(M_{i}^{g*},M_{j}^{g*},F_{i}^{g*},J^{*},I^{*}\right)\geq U_{g}\left(M_{i}^{g},M_{j}^{g},F_{i}^{g},J^{*},I^{*}\right)$$
(17)$$U_{j}\left(G^{*},m_{i}^{j*},f_{i}^{j*},I^{*}\right)\geq U_{j}\left(G^{*},m_{i}^{j},f_{i}^{j},I^{*}\right)$$
(18)$$U_{i}\left(G^{*},J^{*},\gamma_{i}^{l*},\gamma_{i}^{g*},\gamma_{i}^{j*},\gamma_{i}^{r*}\right)\geq U_{i}\left(G^{*},J^{*},\gamma_{i}^{l},\gamma_{i}^{g},\gamma_{i}^{j},\gamma_{i}^{r}\right)$$

## 3. Equilibrium Analysis of Stackelberg Game

### 3.1. Existence Analysis of the Equilibrium Solution

**Theorem** **1.**
*Stackelberg Equilibrium point exists in the Stackelberg multi-layer game G.*


**Proof**. As the leader, the utility of base station G is:
(19)$$\begin{array}{cc} U_{g}\left(M_{i}^{g},M_{j}^{g},F_{i}^{g},J^{*},I^{*}\right) & =P_{g}-C_{g}\\ \; & =\sum_{i\in I}M_{i}^{g}F_{i}^{g}Need_{i}^{g}-\sum_{i\in I}\sum_{j\in J}M_{j}^{g}D_{i}Need_{i}^{j}\gamma_{i}^{r} \end{array}$$
get the partial derivatives respectively, there are:
(20)$$\begin{array}{cc} \; & \frac{\partial U_{g}}{\partial M_{i}^{g}}=F_{i}^{g}Need_{i}^{g}\geq 0\\ \; & \frac{\partial U_{g}}{\partial F_{i}^{g}}=M_{i}^{g}Need_{i}^{g}\geq 0\\ \; & \frac{\partial U_{g}}{\partial M_{j}^{g}}=-D_{i}Need_{i}^{j}\gamma_{i}^{r}\leq 0 \end{array}$$
therefore, $U_{g}$ is monotonous to $M_{i}^{g},M_{j}^{g},F_{i}^{g}$, then $M_{i}^{g},M_{j}^{g},F_{i}^{g}$ are bounded could be deduced according to the constraint condition (4). The condition (16) is satisfied when $M_{i}^{g*}=\max M_{i}^{g},M_{j}^{g*}=\min M_{j}^{g}\;\mathrm{and}\;F_{i}^{g*}=\max F_{i}^{g}$.As the vice-leader, the utility of UAV J is:
(21)$$\begin{array}{cc} U_{j}\left(G^{*},m_{i}^{j},f_{i}^{j},I^{*}\right) & =P_{j}^{compute}+P_{j}^{relay}-C_{j}^{compute}-C_{j}^{trans}\\ \; & =\sum_{i\in I}m_{i}^{j}f_{i}^{j}Role_{i}^{j}+\sum_{i\in I}\sum_{j\in J}M_{j}^{g}D_{i}Need_{i}^{j}\left(1-Role_{i}^{j}\right)\\ \; & -\sum_{i\in I}\frac{\varphi_{i}}{f_{i}^{j}}\eta_{i}Need_{i}^{j}Role_{i}^{j}-\sum_{i\in I}\sum_{j\in J}p_{j}\frac{D_{i}}{Rate_{i}^{j}}Need_{i}^{j}\left(1-Role_{i}^{j}\right) \end{array}$$
get the partial derivatives respectively, there are:
(22)$$\begin{array}{cc} \; & \frac{\partial U_{j}}{\partial m_{i}^{j}}={f_{i}}^{j}Role_{i}^{j}\geq 0\\ \; & \frac{\partial U_{j}}{\partial f_{i}^{j}}=m_{i}^{j}Role_{i}^{j}+\frac{\varphi_{i}}{{\left(f_{i}^{j}\right)}^{2}}\eta_{i}Need_{i}^{j}Role_{i}^{j}\geq 0 \end{array}$$
therefore, $U_{j}$ is monotonous to $m_{i}^{j},f_{i}^{j}$, then $m_{i}^{j},f_{i}^{j}$ are bounded could be deduced according to the constraint condition (10). Let $m_{i}^{j}=\max m_{i}^{j},f_{i}^{j}=\max f_{i}^{j}$, the condition (17) is satisfied.As follower, the utility of user I is:
(23)$$\begin{array}{cc} U_{i}\left(G^{*},J^{*},\gamma_{i}^{l},\gamma_{i}^{g},\gamma_{i}^{j},\gamma_{i}^{r}\right) & =\left[\begin{array}{cccc} \gamma_{i}^{l} & \gamma_{i}^{g} & \gamma_{i}^{u} & \gamma_{i}^{r} \end{array}\right]\cdot \left(-C_{i}^{compute}-C_{i}^{pay}\right) \end{array}$$
since $C_{i}^{compute}$ and $C_{i}^{pay}$ are constant for $\gamma_{i}$, $U_{i}$ could be simplified as:
(24)$$U_{i}\left(G^{*},J^{*},\gamma_{i}^{l},\gamma_{i}^{g},\gamma_{i}^{j},\gamma_{i}^{r}\right)=C_{1}\gamma_{i}^{l}+C_{2}\gamma_{i}^{g}+C_{3}\gamma_{i}^{j}+C_{4}\gamma_{i}^{r}$$
therefore, $U_{i}$ is linearly related to $\gamma_{i}$, and the game of this layer is a linear programming problem. The constraint given by (14) is a convex set, then the objective function $U_{i}$ obtains the optimal value at the vertex of the feasible domain [29].In summary, the Stackelberg Equilibrium point exists in multi-layer game G. □

### 3.2. Equilibrium Uniqueness Analysis of Stackelberg Game

**Theorem** **2.**
*The stackelberg game equilibrium point is unique.*


**Proof**. For the leader and vice-leader layer, it could be known from (20) and (22) that $U_{g}$ and $U_{j}$ are monotonic to $M_{i}^{g},M_{i}^{j},F_{i}^{g}$ and $m_{i}^{j},f_{i}^{j}$. Therefore, the optimal solutions obtained are unique.For follower layer, according to the constraint (14), the feasible domain of $\gamma_{i}^{l},\gamma_{i}^{g},\gamma_{i}^{j},\gamma_{i}^{r}$ are convex sets, and the optimal solution is obtained at the vertex. In addition, the feasible domain formed by constraint (14) is a pyramid, so the vertex is unique, and the optimal solution of the follower layer is unique [29].In summary, the Stackelberg Equilibrium point in multi-layer game G is unique. □

## 4. Stackelberg Game Resource Pricing and Computation Allocation Algorithm

Parameters initialization, initialize the values of the parameters in the simulation environment and the strategy for the first round of the game. Get the distances between each base station, UAV and user. And users will choose to establish a connection with the nearest base station or UAV in the initial state.Resource pricing and allocation, the players in each layer play a game of resource allocation and pricing, updating the optimal strategy based on the current round in the order of leader, vice-leader, and follower, $M_{i}^{g},M_{j}^{g},m_{i}^{j},F_{i}^{g},f_{i}^{j},\gamma_{i}^{l},\gamma_{i}^{g},\gamma_{i}^{j},\gamma_{i}^{r}$ are involved (as in Algorithm 1).Repeat step (2) for iterations, and eventually converge to equilibrium through multiple rounds of the game.

**Algorithm 1** Stackelberg game resource pricing and allocation algorithm **Input:** Location: bs,uav,user; Computation amount: $D_{i}$; Iteration number: epoch  **Output:** Price: $M_{i}^{g},M_{j}^{g},m_{i}^{j}$; Resource amount: $F_{i}^{g},f_{i}^{j}$; Offload rate:$\gamma_{i}^{l},\gamma_{i}^{g},\gamma_{i}^{j},\gamma_{i}^{r}$1: Simulation parameter initialized according to the distance relationships and computation amount.2: **for** episode = 1 to epoch **do**3:  **for** b = 1 to $bs_{num}$ **do**4:   $F_{i}^{g}$=$D_{i}\cdot \left(\gamma_{i}^{g}+\gamma_{i}^{r}\right)$;5:   **if**$\gamma_{i}^{g}\geq 0$ and $Need_{i}^{g}$= =1 **then**6:    **if**
$\min M_{i}^{g}\leq 2M_{i}^{g}\leq \max M_{i}^{g}$
**then**7:      $M_{i}^{g}$=$2\cdot M_{i}^{g}$; $M_{i,low}^{g}$=$M_{i}^{g}$;8:   **else**9:    $M_{i,high}^{g}$=$M_{i}^{g}$; $M_{i}^{g}$=$\frac{M_{i,high}^{g}+M_{i,low}^{g}}{2}$;10:   **if**$\gamma_{i}^{r}\geq 0$ and $Need_{i}^{g}$ = = 1 **then**11:    $M_{j,high}^{g}$=$M_{j}^{g}$; $M_{j}^{g}$=$\frac{M_{j,high}^{g}+M_{j,low}^{g}}{2}$12:   **else**13:    **if**
$\min M_{j}^{g}\leq 2M_{j}^{g}\leq \max M_{j}^{g}$
**then**14:      $M_{j,low}^{g}$=$M_{j}^{g}$; $M_{j}^{g}$=$2\cdot M_{j}^{g}$15:  **for** u = 1 to $uav_{num}$ **do**16:   $f_{i}^{j}$=$D_{i}\cdot \gamma_{i}^{j}$17:   **if**
$\gamma_{i}^{j}\geq 0$ and $Need_{i}^{j}$= =1 **then**18:    **if**
$\min m_{i}^{g}\leq 2m_{i}^{g}\leq \max m_{i}^{g}$
**then**19:      $m_{i}^{g}$=$2\cdot m_{i}^{g}$; $m_{i,low}^{g}$=$m_{i}^{g}$;20:   **else**21:    $m_{i,high}^{g}$=$m_{i}^{g}$; $m_{i}^{g}$=$\frac{m_{i,high}^{g}+m_{i,low}^{g}}{2}$;22:  **for** i = 1 to $user_{num}$ **do**23:   update $\gamma_{i}^{l},\gamma_{i}^{g},\gamma_{i}^{j},\gamma_{i}^{r}$ by utility function according to $M_{i}^{g},M_{j}^{g},m_{i}^{j},F_{i}^{g},f_{i}^{j}$;24:   update $Need_{i}^{g},Need_{i}^{j}$ according to $\gamma_{i}^{l},\gamma_{i}^{g},\gamma_{i}^{j},\gamma_{i}^{r}$.25:  record the utility of each player in this iteration.26: **return**$M_{i}^{g},M_{j}^{g},m_{i}^{j},F_{i}^{g},f_{i}^{j},\gamma_{i}^{l},\gamma_{i}^{g},\gamma_{i}^{j},\gamma_{i}^{r}$ and the final utility of each players.

## 5. Results

### 5.1. Simulation Parameter Setting

The basic simulation environment is a MEC network composed of two base stations, 6 UAVs, and eight users. The locations of the base stations, UAVs, and users (randomly generated) are distributed in a Cartesian plane coordinate system as shown in Figure 2. The computing resources of base stations, UAVs, and users are all 64 GB, 8 GB, and 32 MB, respectively. In addition, the computation amount of users are random in 10 MB∼1 GB. Other parameters are shown in Table 1. The program design and result diagram of the simulation are completed in MATLAB, and the operating system is Windows 10, 64-bit professional version. The base stations, UAVs, and users in the simulation scene are virtual devices imitating real device parameters.

### 5.2. Simulation Result and Performance Comparison

Figure 3 shows the iteration of the three roles of the base station, UAV, and user in the Stackelberg multi-layer game. It can be found that after about 250 iterations, the income or overheads of players at each layer finally reached the equilibrium state, indicating that the algorithm can achieve Stackelberg equilibrium. During the whole process, as shown in Figure 3a, the incomes of BS1 and BS2 were basically stable, which changed suddenly at about 150 iterations and returned to equilibrium after about 50 iterations. In Figure 3b, the overhead of user2 changed dramatically about 150 iterations, indicating that user2 switched the offloading object. And after about 50 iterations, user2 reached the equilibrium state. The overheads of other users gradually increase in the oscillation and enter the equilibrium state when they reach the certain values, which indicates that the prices for computation services had reached the maximum value, and the offloading objects had not changed during that period. In Figure 3c, the incomes of UAVs fluctuated greatly before convergence, indicating that the competition of UAVs in the game of resource allocation was fierce. We can also find that after about 150 iterations, the incomes of UAVs were oscillations in several points, but as they gradually entered the equilibrium state, there were fewer and fewer oscillations until they reach the equilibrium and no longer oscillate. The specific computation offloading situation in Stackelberg equilibrium is shown in Figure 4.

The benchmark algorithm is RANDOM (the users decide the offload objects and proportions randomly) and LOCAL (all users execute the task locally), and the total profit of a multi-level game is used as the performance evaluation standard. The total profit of the system can be expressed as:(25)$$Utility=U_{g}+U_{j}-U_{i}$$

As shown in Figure 5, we compare the total system profits of the Stackelberg game algorithm (SGA) with RANDOM and LOCAL. Firstly, it can be found that the SGA strategy reached an equilibrium state after about 250 iterations. Secondly, the total system profit of SGA in the equilibrium state is about 80% more than that of random strategy, and the profits of SGA and RANDOM are more than LOCAL.

We also compare the SGA in an equilibrium state with the RANDOM and LOCAL strategy in five different scenarios, as shown in Figure 6. Figure 6a shows the total system profit in the scenarios where the quantity of users is 2, 4, 8, 12, and 16, respectively. We can see SGA is significantly better than RANDOM and LOCAL in all five scenarios, and the total profit of the system increases with the quantity of users. Figure 6b is the performance comparison chart of SGA, LOCAL, and RANDOM in scenarios with different user positions, and the coordinates of users in the other four scenarios are also randomly generated. It can be found that the SGA algorithm is superior to RANDOM and LOCAL in these scenarios. In addition, we can find that the total system profit of RANDOM fluctuates greatly, and the profits in some scenarios are not as good as the LOCAL. While the total system profit of SGA is more stable. Figure 6c is the performance comparison chart of 5 scenarios with different quantities of user computation. They are all randomly generated, with the value of (673,978,768,843,408,616,543,424), (971,609,720,303,460,49,386,362), (288,817,451,807,791,283,169,255), (638,425,906,418,255,540,938,661), (395,259,848,946,377,468, 482,576) MB, respectively. It can be found that SGA is better than RANDOM and LOCAL. Since the amount of computation directly affects the overhead, it is normal for the total system profit to fluctuate.

## 6. Conclusions

This study aims to solve the problem of computing resource management in a multi-UAV-enabled edge computing scenario. According to the different interests of base stations, UAVs and users, we construct a three-layer Stackelberg game model by comprehensively considering computing latency, energy consumption, and specific profit. The equilibrium state is achieved after several iterations and verified by simulation. The Stackelberg game algorithm has better total system profit in multiple different scenarios and has certain robustness in the equilibrium state than the random strategy. Therefore, in the future scenario of multi-UAV-enabled edge computing, Stackelberg game theory has application prospects in solving complex resource problems. 

## Figures and Tables

**Figure 1 sensors-22-03854-f001:**
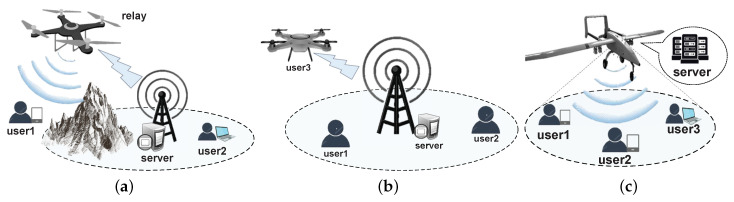
The roles of UAVs in computing resource allocation for MEC networks. (**a**) uav as relay. (**b**) uav as user. (**c**) uav as server.

**Figure 2 sensors-22-03854-f002:**
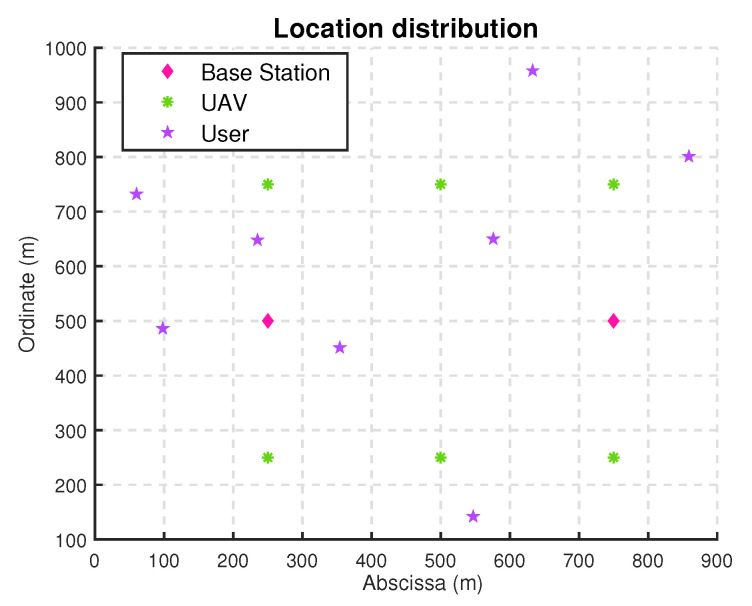
Location distribution in simulation (UAVs’ coordinates are taken from ground projection).

**Figure 3 sensors-22-03854-f003:**
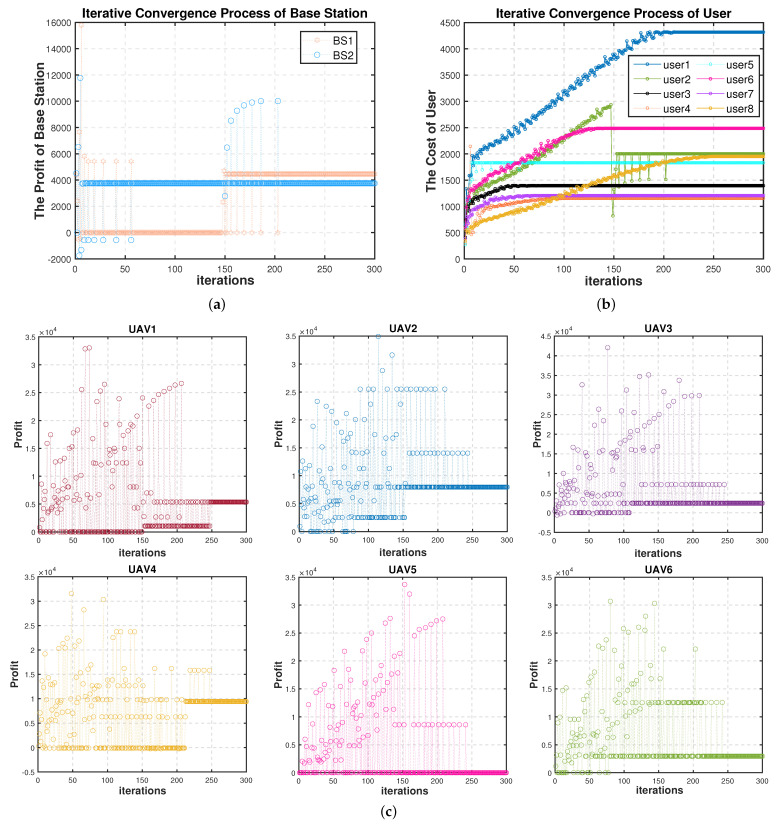
Iteration process in each layer. (**a**) iteration process of base stations. (**b**) iteration process of users. (**c**) iteration process of UAVs.

**Figure 4 sensors-22-03854-f004:**
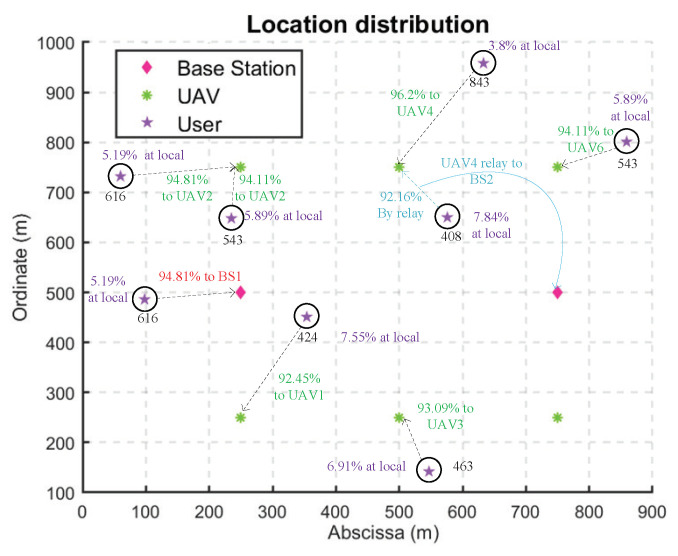
Optimal computation offload strategy.

**Figure 5 sensors-22-03854-f005:**
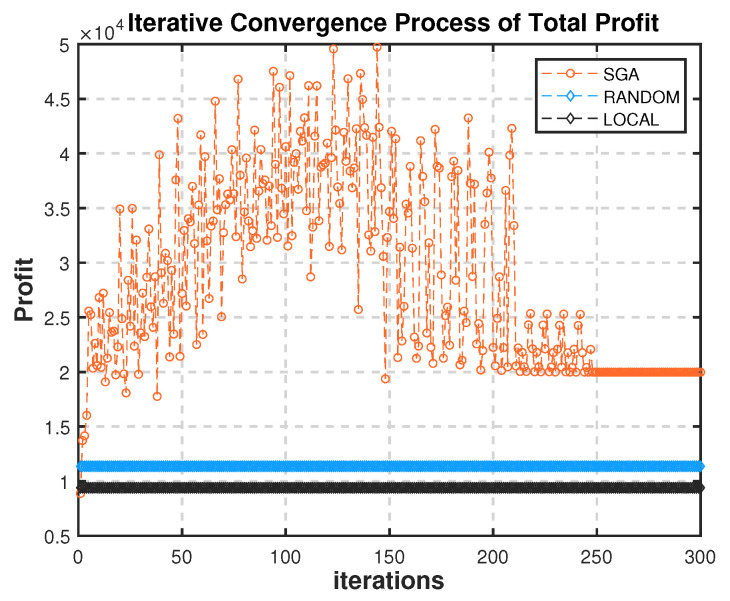
Iterative process of total profit of the system and comparison with random strategy.

**Figure 6 sensors-22-03854-f006:**
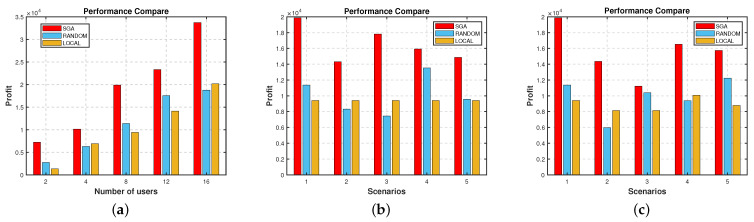
Performance comparison in different scenarios. (**a**) Different number of users. (**b**) Different user locations. (**c**) Different amount of computation.

**Table 1 sensors-22-03854-t001:** Other simulation parameter settings.

**Parameter**	**Symbolic Representation**	**Value**
Energy consumption per CPU cycle	$\eta_{i}$	8
Weights about local computation	$\alpha_{local,i}^{time}\alpha_{local,i}^{energy}$	(0.5, 0.5)
Weights about offload to BS	$\alpha_{g,i}^{time},\alpha_{g,i}^{energy},\alpha_{g,i}^{pay}$	(0.3, 0.3, 0.4)
Weights about offload to UAV	$\alpha_{j,i}^{time},\alpha_{j,i}^{energy},\alpha_{j,i}^{pay}$	(0.3, 0.3, 0.4)
Weights about relay	$\alpha_{relay,i}^{time},\alpha_{relay,i}^{energy},\alpha_{relay,i}^{pay}$	(0.3, 0.3, 0.4)

## Data Availability

Not applicable.

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
