# Peer review of "Computation Offloading in UAV-Enabled Edge Computing: A Stackelberg Game Approach"

_sensors, 2022, doi:10.3390/s22103854_

Round 1

Reviewer 1 Report

According to the interests of three different roles: base station, UAV, and user, the authors comprehensively consider the factors such as time delay, operation, and transmission energy consumption in a multi-layer game to improve the overall system performance. 

The authors should further improve the submission based on the following comments:

  1. There are too many symbols in the manuscript. For better understanding, the authors should summarize them all in one table.
  2. A concrete example from real world is necessary to be added for better describing the research motivation in the paper.
  3. More experiment details are needed in the evaluation part, e.g., experiment software and hardware configurations, experiment repetition times and so on.
  4. Big data challenge in MEC has been studied for long time. Therefore, I suggest the authors to introduce the following related literatures published in recent two years: Utility Aware Offloading for Mobile-Edge Computing; Big Data with Cloud Computing: Discussions and Challenges; Artificial intelligence for edge service optimization in internet of vehicles: A survey; A Survey on Algorithms for Intelligent Computing and Smart City Applications.
  5. Proofread the whole paper and correct the existing grammar and syntax mistakes.

Author Response

Thank you very much for you comments, please see the attachment.

Reviewer 2 Report

Line 53 - Typo needing correction "sloved"

Lines 138 - 141 - Please give detail about the distance influence in the choices to be made.

Figure 2 - The X and Y-axis have no units.

Figure 3 - Typos needing correction "iteartions"

REgarding Figure 5-c) please provide detail about the quantities of user computation considered for the five different scenarios (in the text).

Author Response

(The authors gave the same response as above.)

Reviewer 3 Report

• Please add a Figure or Table about the optimal structure of the proposed method. In addition, provide the values of all parameters of the proposed method in the table. 
• Please specify how the parameters of the proposed method were selected. 
• Please specify if the parameters of the proposed method were optimized. If so, please write how the proposed parameters were optimized?

Author Response

(The authors gave the same response as above.)

Reviewer 4 Report

  1. This paper proposed a multi-layer Stackelberg game model to solve the computation offloading in a multi-UAV environment. Since multi-UAV issues and the multi-layer Stackelberg game model have been studied in the literature,  the contributions of this study need more information to emphasize its novelty.
  2. More literature surveys and comparisons of this work with previous works are required (such as Two-Layer Stackelberg Game-Based Offloading Strategy for Mobile Edge Computing).
  3. How is the communication cost considered in this work? Communication is essential and needs more information to allocate the communication resource, such as SINR.
  4. It is unclear how latency and energy constraints are considered in the model?
  5. Given that there are many offloading strategies of similar work, it is necessary to compare with the state-of-the-art results, not with the random strategy in Fig. 5.

Author Response

(The authors gave the same response as above.)

Round 2

Reviewer 3 Report

Authors mentioned all my comments and paper can be accepted

Author Response

(The authors gave the same response as above.)

Reviewer 4 Report

  1. How is the communication cost considered in this work? Here, communication refers to data transmission required for offloading a task. It is especially important since the communication models of UE to the base station and UE to UAV are very different in terms of transmission bandwidth, and energy consumption. Without Channel models for these transmission scenarios, it is impossible to estimate the transmission latency and energy consumption.
  2. Since latency and energy were considered in some previous works, adopting their work in this work's simulation is possible for comparison purposes.

Author Response

(The authors gave the same response as above.)

Round 3

Reviewer 4 Report

It is easy to have a notation Rate_{i}^{j,g} to denote the transmission rate of the UAV link, but it is not realistic to calculate this value without a given channel model. In general, it is not a fixed value.